# Multi-Energy and Fast-Convergence Iterative Reconstruction Algorithm for Organic Material Identification Using X-ray Computed Tomography

**DOI:** 10.3390/ma16041654

**Published:** 2023-02-16

**Authors:** Mihai Iovea, Andrei Stanciulescu, Edward Hermann, Marian Neagu, Octavian G. Duliu

**Affiliations:** 1Accent Pro 2000 srl, 25A, Mărășești Str., 077125 Magurele (Ilfov), Romania; 2Department of Structure of Matter, Earth and Atmospheric Physics, Astrophysics, Faculty of Physics, University of Bucharest, 405, Atomistilor Str., 077125 Magurele (Ilfov), Romania; 3Geological Institute of Romania, 1, Caransebes Str., 012271 Bucharest, Romania

**Keywords:** X-ray multi-energy computed tomography, algorithm, confident identification, organic material, density, effective atomic number

## Abstract

In order to significantly reduce the computing time while, at the same time, keeping the accuracy and precision when determining the local values of the density and effective atomic number necessary for identifying various organic material, including explosives and narcotics, a specialized multi-stage procedure based on a multi-energy computed tomography investigation within the 20–160 keV domain was elaborated. It consisted of a compensation for beam hardening and other non-linear effects that affect the energy dependency of the linear attenuation coefficient (LAC) in the chosen energy domain, followed by a 3D fast reconstruction algorithm capable of reconstructing the local LAC values for 64 energy values from 19.8 to 158.4 keV, and, finally, the creation of a set of algorithms permitting the simultaneous determination of the density and effective atomic number of the investigated materials. This enabled determining both the density and effective atomic number of complex objects in approximately 24 s, with an accuracy and precision of less than 3%, which is a significantly better performance with respect to the reported literature values.

## 1. Introduction

The uninterrupted development of international trade, besides the beneficent effects related to market globalization, has encountered a negative influence due to the increased use of illicit drugs and explosive trafficking, which have proved serious problems for law enforcement agencies, anti-narcotic police, and customs on border management. As the problem of fast and confident explosive detection represents one of the most fundamental aspects of passenger security, until present, a multitude of non-invasive techniques were proposed that were mainly based on the differential attenuation or diffraction of nuclear radiation. It is the case of X-ray single [1,2], dual [3,4], phase contrast [5], multiple-energy [6,7], diffraction [8], or even neutron computed tomography (CT) [9]. In addition to these techniques, Raman [10] or nuclear quadrupole resonance [11] spectroscopy as well as infrared photothermal imaging [12] gave remarkable results in this field, just to mention some alternative methods not involving the use of X-rays.

Different from the medical applications of dual and multiple-energy CT, which produce more precise images necessary for a better diagnostic [13,14], the forensic use of dual and multiple CT permits the simultaneous determination of the effective atomic number Z_*eff*_ and density ρ of controlled material, accurately differentiating explosives from the other organic materials in the luggage. In this regard, it is worth mentioning that densities or effective atomic numbers alone cannot differentiate explosives; this can only be achieved if they are simultaneously used [6,15].

Therefore, there is a strong motivation for the development of automated and non-invasive systems used to control the content of luggage or different parcels before being admitted to shipping or selected for manual examination. This control should be fast and reliable given the increasing volume of air, land, and sea traffic. In this regard, it is worth mentioning that the screening is not without errors, but the reason for its use is to reduce to an absolute minimum the number of false positive or false negative results.

The development of X and gamma-ray solid-state spectrometric detectors such as CdTe or Cd(Zn)Te [16,17], which are able to record the energy spectrum of incident radiation on a large domain of energies subdivided into numerous bins, has stimulated the development of the multi-spectral CT (MSCT) [4,6,18,19,20]. This new variant of classical dual-energy CT (DECT) has permitted determining the entire energy spectrum of linear attenuation coefficients (LACs), which, given their complex dependency on photon energy and the atomic number, have shown to be extremely useful in identifying diverse materials, even if their ρ and Z_*eff*_ are close.

In the case of DECT, for each voxel of the considered section, there are two different values of the LAC corresponding to the two involved energies necessary to simultaneously determine Z_*eff*_ and ρ. On the contrary, the use of multiple energy has presented the advantage of a better resolution concerning Z_*eff*_ and ρ, which significantly increases the ability of MSCT to differentiate elements whose parameters are relatively closer. On the other hand, the number of photons per energy bin decreases with increasing the bin numbers, which, in turn, increases the quantum noise. To maintain a good spatial and discriminant resolution by keeping, at the same time, both the acquisition and processing time reasonably low, more techniques were elaborated to reduce the number of projections [21,22] or to find new algorithms [6,23,24].

In this regard, the use of neural networks (NNs) has brought impressive improvements for object and material identification [25,26]. Once well trained, a neural network can, in this particular case of CT, significantly increase the image quality by reducing the noise, optimizing the contrast, or better evidencing the contours [27,28]. To accomplish this task, the NN should be trained by processing a significant number of images, which, in our case, could be accomplished by using more samples consisting of known materials, such as explosive simulants, cosmetics, various organic compounds, etc.

In spite of remarkable performances in identifying a large category of materials, due to the simultaneous use of a great number of energy bins, the processing time sometimes exceeds a reasonable value, thus reducing the number of objects that can be examined in a considered time interval.

Under these circumstances, the main task of this project consisted of the elaboration of a novel, faster MSCT volumetric reconstruction method that serves as the basis for a rapid and non-invasive system to evidence the presence of prohibited materials, especially explosives, based on the real-time determination of both Z_*eff*_ and ρ values of controlled items.

In order to perform this task, it was necessary to: i. elaborate a procedure to compensate for the beam hardening and other effects that may alter the energy dependency of the LAC in the domain of 20–160 keV used for X-ray CT; ii. develop a 3D reconstruction method that can deliver the accurate results needed for material identification while using a reduced number of projections and having a low computation time, such that it can be used in a production environment; iii. refine the state-of-art iterative reconstruction algorithms presented in [21,23] to ensure a faster convergence that permits, by analysing the corrected LAC multi-energy spectra, the simultaneous determination of both Z_*eff*_ and ρ values.

It is worth mentioning that, for a higher accuracy and precision, we used the LAC published by the National Institute of Sciences and Technologies (NIST) https://physics.nist.gov/PhysRefData/XrayMassCoef/tab4.html (accessed on 12 January 2023).

The results of this project will be further presented and discussed.

## 2. Materials and Methods

### 2.1. Samples

To obtain more data necessary to elaborate a reliable set of correction, we have constructed five staircases shaped test-pieces made of Certified Reference Material (CRM) Polyethylene terephthalate (PET), Polytetrafluoroethylene (PTFE), Polyethylene (PE), and Polyvinyl-chloride (PVC), as well as Al. PE, PTFE, PER, PBC, and PVC thicknesses ranged from 10 mm to 100 mm in 10 mm steps, while the Al sample have only six steps of which thickness varied between 2.8 and 51.9 mm (Table 1). The accuracy and precision of our algorithms was checked using the explosive simulants PA12, XM03X, XM04X1, and XM08X, all of them provided by XM Materials http://www.xm-materials.com/xray_equivalency.html (accessed on 12 January 2023).

### 2.2. Computed Tomograph

We have used a home-made Computed Tomograph (CT) provided with a tungsten cathode microfocus Gilardoni AION X-ray source (Gilardoni S.p.A.), https://www.gilardoni.it/en/company/ (accessed on 10 January 2023) operating under 150 kV acceleration voltage and 0.5 mA current and a classic transmission set-up (Figure 1a). Samples were placed on a rotary table permitting also to adjust their vertical position with respect to the X-ray fan beam, so that to scan different slices. The distances source-rotation center and source-detector were of 315 mm and, respectively, of 687 mm, assuring a magnification of approximately 2.2. Five in-line MultiX ME-100TM detectors, (Detection Technology Plc., Oulu, Finland), https://www.deetee.com/company/ (accessed on 10 January 2023), each of them consisting of 128 pixels of 0.8 × 0.8 mm^2^ configured by manufacturer with 64 equal-energy bins of 2.2 keV, were used (Figure 1b,c). This arrangement covering an energy range from 20 keV to 150 keV matched the entire generator energy spectrum, permitting a simultaneous acquisition in 23 min of 64 sets of 36 projections corresponding to each energy bin. Under these circumstances, finally resulted 640 × 4000 pixels images at an optical resolution of 64 dpi. The chosen number of projections represented a trade off between the quality of reconstructed images and acquisition time, so that the real error in determining both Z_*eff*_ and ρ to be less than 3%.

Data acquisition was performed using a Detector Technology Plc. software while LabVIEWTM programs (National Instruments, Austin, TX, USA), https://www.ni.com/ro-ro.html (accessed on 12 January 2023) was used to adjust the sample vertical positioning, perform the incremental rotations, as well as other operations.

### 2.3. Linear Attenuation Coefficient Corrections

As mentioned before, given the complex dependence of LAC on atomic number and photons energy, as well as different perturbative effects which imply the interaction of incident X-ray with sample and detectors, the LAC energy dependency different from the expected ones as those provided by NIST. In this regard, potential cross-talk and charge sharing between neighbouring detectors, pulse pile-up or incomplete charge collecting together with the Compton scattering and local X-ray fluorescence radiation at the detectors level or beam hardening effect [29] at the sample level make the attenuation no longer obeying an exponential decay function of material thickness.

To correct these, using the NIST data, we have calculated, for each segment of test-sample, 64 values of the LAC following the exponential attenuation law: (1)$$\mu_{ij}=\frac{1}{l}n\frac{I_{0}}{I_{ij}}$$
where $\mu_{ij}$ represents the LAC corresponding to object length $x_{i}$ and energy *j* for each bin between 20 to 160 keV, $I_{0}$ and $I_{ij}$ represents the intensity of incident and emergent photons beam as measured on transmitting geometry.

The influence of scanned area on LAC values was reduced by manually selecting the region of interest for each image.

Due to the fact that resulted LAC experimental spectra showed significant differences with respect to the NIST ones, we have calculated for each energy bin and each material thickness a correction coefficient:(2)$$d_{ij}=\frac{\mu_{NIST,i,j}}{\mu_{meas,i,j}}$$
where $\mu_{NIST,i,j}$ and $\mu_{meas,i,j}$ represents the experimental and, respectively, the NIST values of the considered material thickness *i* and energy *j*.

The calibration was executed using the LAC of PET, PTFE, and PE, while all five samples were used to evaluate the correction quality. Further, by using a smoothing spline interpolation, we have calculated for each energy bin a correction set of values to be used for further scans. The resulting spline curve was resampled at a smaller rate to reduce the computing time. As a result, the energy dependence of LAC has gained a remarkable resemblance with the NIST data, especially in the case of PET, PTFE, and PE, as the bi-plots reproduced in Figure A1 and Figure A2 illustrate. These observations were confirmed by an ANOVA multi-sample Mann–Whitney which has shown that the best fit was realized for materials consisting of light elements H, C, N, and F, while the presence of heavier Al (Z = 13) and especially Cl (Z = 17) reduced the probability the corrected dependencies to be closer to NIST values, which according to (Table A1) was zero. In our opinion, this result reflects the predominance of photoelectric in the case of higher atomic number elements at X-ray energies below 35–40 keV (Figure A1 and Figure A2).

All data calibration and correction steps were performed using the MATLAB environment, using the existing data fitting tools while statistical analysis was performed by PAST 4.09 freeware [30].

## 3. Materials Reconstruction

The second stage of this project consisted of developing fast 3D reconstruction method needed for an accurate material identification based on a reduced number of projections.

In this regard, the inverse problem of finding the volumetric reconstruction given a set of CT projections consists in determining the optimal values of the voxels volume such that the error between the forward projection through the volume and the acquired sinograms reaches a minimum. Depending on the number of projections and preliminary knowledge on the reconstructed volume, there are multiple ways of approaching this problem. Our proposed implementation relies on an iterative method for approximating the voxel values based on the forward projection error. Using an adjustable scaling factor for the data corrections, as well as an adaptive threshold based on the reconstruction stage, we were able to achieve convergence with considerably fewer iterations compared to similar algorithms. Since we are working with multi-spectral images, our method will also include a data regularization routine for using information from other energy bands, based on the gradient’s correlation across the acquired energy spectrum.

### 3.1. Mathematical Formulation

Let $u\in R^{M\times N}$, where $u_{m,n}$ is the *m*-th pixel from the reconstructed image corresponding to the *n*-th energy bin, *A* represents the forward projection operator through *u*, while $p\in R^{N\times K}$, where $p_{n}$ is the sinogram corresponding to the *i*-th energy bin. The data fitting problem of reconstructing the images aims to minimize the error ${∥Au-p∥}^{2}$.

However, this type of solver reconstructs each image separately, not taking advantage of the known similarities that exist between the images at different energies. To address this issue, we introduce a data regularization scheme $L_{\infty }-VTV$ to the mathematical formulation of the problem. This regularization parameter represents the sum of the maximum of the gradients of a multiple channel image. This norm correlates the gradients strongly, while allowing some outliers. A weighting parameter is used to balance the two terms when solving the minimization problem and represents the trade-off between the data misfit term and the regularization term [31].

The reconstruction problem can now be expressed as:(3)$$\min_{u⩾0}\lambda \frac{1}{2}{∥Au-p∥}^{2}+\sum_{m=1}^{M}\left[\max_{1⩽i⩽N}\frac{\partial }{\partial x}u_{i,m}+\max_{1⩽i⩽N}\frac{\partial }{\partial y}u_{i,m}\right]$$

### 3.2. Reconstruction Implementation

In practice, we are using an iterative algorithm to approximate the solution to the minimization problem. Prior to the reconstruction phase, the scanning system’s geometry was calculated, so we know through which voxel each X-ray passes, as well as the corresponding path length. The number of rays was calculated as the detector pixels number multiplied by the number of projections. In this arrangement it was necessary to compute the geometry once, since it is the same for all reconstructions. Each voxel has a size of 2 × 2 mm^2^, a parameter which can be set in the geometry generation algorithm. To calculate the scanning setup’s geometry, the reconstruction area was divided into equally sized squares, defined by a grid of equal spaced parallel and perpendicular lines. By knowing the coordinates of the X-ray source and of every detector pixel, we can determine where the ray intersects each of the lines defining our reconstruction grid.

From these coordinates, we can then determine the pixel indices intersected by a specific ray and the ray length through it (Figure 2). This path tracing method is an implementation adapted from Siddon’s algorithm [32]. Given the nature of this algorithm, it is easily scaled to benefit from multiple processor threads, thus reducing the computing time and taking full advantage of available resources. In our application, however, the geometry data can be computed in advance, and it need only to be loaded from disk for the reconstruction step. However, the geometry parameters, such as source to detector distances, detector panel angles, and spacing, need to be finely adjusted to account for measurement and construction errors in the physical setup.

### 3.3. Direct Data Fitting Term

For each reconstruction iteration, we have iterated through the array of computed rays *r*, and for each ray *r_i_*, we have calculated the forward projection through that ray, and compared it with the corresponding sinogram pixel $p_{n,i}$, *n* being the current energy bin. The correction thus calculated were distributed to all pixels the ray passes through, according to each pixel’s weight. The weights are either constant for all pixels or depend on the ray’s length through them. The decisive factor in calculating these weights is how far we are in the reconstruction process. Accordingly, the first two iterations gave constant weights, then they will be changed for better data fidelity. This process, independently repeated for each energy bin, represents the first part of the minimization problem.

Since the process is identical and independent for all energy bins, further we will restrain to one energy bin. Let $l_{m}=1.13l$ be the length of the *m*-th ray through *m*-th voxel $\mu_{m}$ the LAC of the *m*-th pixel and *L* the total length of the ray through the object. The sinogram value corresponding to ray $r_{i}$ is *u_i_*. The forward projection through the ray is $\mu_{total}=\sum_{m=1}^{N}l_{m}\cdot \mu_{m}$, *N* being the number of voxels crossed by the *m*-th ray while the error $e_{i}=p_{i}-\mu_{total}$ is such that the corrected value for the m pixel becomes: $\mu_{m}^{q+1}=u_{m}^{q}+\frac{k\cdot e\cdot l_{m}}{L}$. Here, the value *k* represents the gain/relaxation factor which is used to prevent overshoots too big in the process estimation. In our implementation, the amplification factor decreases from 1.5 to 1.0 over the all 11 iterations. This allows significant changes at the process beginning followed by a reduction in corrections magnitude, ensuring in this way the convergence. Before updating the reconstructed image, an adaptive threshold was applied to the modified pixels to eliminate values which are too small. The threshold is calculated as a lower percentage of the range of the pixels’ values. The percentage is changed every few iterations and is defined at run time, just like the amplification factor.

In this regard, as this part of the reconstruction is performed independently for each energy bin, the computations can be performed in parallel to increase performance. In our experiments, the reconstruction time was reduced between 9 and 10 times when running the reconstruction in parallel, rather than doing them sequentially. This improvement was observed on a consumer personal computer, so the use of a high-performance workstation may not be necessary.

### 3.4. Data Regularization Term

After the independent data fitting step is over, the images for every energy bin are processed together by a regularization algorithm. The best results were obtained using the Duran’s implementation of the $L_{m}-VTV$ algorithm [33] of which ANSI C source code is provided either.

The algorithm computes the image gradients in the *x* and *y* directions for each energy bin independently, then penalizes the maximal gradients difference across the spectrum by adjusting the pixel values to decrease the gradient value in that spot. In this way, the high9 gradient discrepancies are adjusted such that the multi-spectral images present a strong coupling between gradients. This procedure permits smaller gradients to be sensitive to energy bins, which is desirable, since materials interact differently with incident rays at different energies.

## 4. Results

The reconstruction algorithm was implemented in C++, configuring the data fitting computation to run in parallel, one thread for each energy bin image. In our test, we achieved convergence in roughly 15 iterations, so that any further iterations did not produce significant improvements (Figure 3a).

As mentioned before, the first four iterations determined the most important reduction in the correction factor. However, even though there are no major changes in image pixels after the fourth iteration, they should continue to refine object contours. Figure 3b presents the corresponding reconstruction after 11 iterations. To illustrate the capability of our algorithm to produce clear margins on interfaces between materials, we have scanned a sample consists of two stair-shaped objects of different materials joined together (Figure 3b).

The procedure showed to be very fast. Usually, for 64 sinograms, each of them with a width of 640 pixels and a height of 36 pixel which correspond to 36 projections finally it resulted 64 reconstructed images of 300 × 300 pixels, one image for each energy bin. Below we provide the program execution metrics averaged over 40 runs with input and output images of the same size as mentioned above. The tests were performed on a computer equipped with an AMD Ryzen 5 2600 CPU (Advanced Micro Devices, Inc., Santa Clara, California, US), https://www.amd.com/en (accessed on 14 February 2023) and 32 GB of 3000 MHz DDR4 RAM resulted in a 23.3 s total execution time (Table 1).

## 5. Discussion

### 5.1. System Calibration

As the final goal of this study consisted of material identification with as low as possible false results, we have used a simplified equation which describes the energy dependence of LAC μ(E) upon the Z_*eff*_ and ρ of a specific material [34]. In this regard, Equation (Equation 4) describes the relation between the LAC and the material properties by considering only photoelectric and Compton incoherent scattering:(4)$$\mu_{\left(E\right)}=p\left[Z_{eff}^{n-1}\cdot p\left(E\right)+c\left(E\right)\right]$$
where: p(E) and c(E) are two parameters which quantify the contribution of photoelectric and Compton effects, and of which values depends only on the photon energy while 3 < *n* < 4.

To check the program performances, we have selected, as mentioned before, 5 CRM with known Z_*eff*_ as well as ρ values for calibration (Table 1) and 5 other different materials for validation (Table 2).

For calibration, we have determined the experimental LAC values for each material and for each energy bin on the reconstructed CT images, by following the procedure previously described. Accordingly, the reconstruction for each material started from a stack of its 64 single channel images, each of them with a size of 300 × 300 pixels. An automated process of thresholding and erosion was used to isolate the Region of Interest (RoI), which once selected was used for all images corresponding to each of the 64 energies. The erosion step was necessary to select the relevant attenuation coefficients, since the object bordering pixels have lower values. Further, for each CRM and each image, the pixel values were averaged inside of each ROI to estimate the LAC average values. Finally, this procedure generated a 5 × 64 LAC attenuation matrix.

The LAC attenuation matrix were used to calculate, using a least squares approximation, the numerical values of the p(E) and c(E) parameters of which values were further considered identical for all materials. Thus obtained values of p(E) and c(E) were used to determine both Z_*eff*_ and ρ values of the other 5 elements based also on the reconstructed CT images, as in the case of CRMs (Table 3).

Fore an easier matrix representation, the LAC were rewritten as:(5)$$\mu_{m}{,}_{k}=p_{m}\left[Z_{n-1}^{m}\cdot p_{k}+c_{k}\right]$$
where *m* represents the material index whereas *k* stays the energy bin index, by considering *M* materials and *K* energy bins.

Computing the optimal *p* and *c* arrays means minimizing the following objective function:(6)$$\sum_{k=1}^{K}\sum_{m=1}^{M}\left[\mu_{m}{,}_{k}-\rho_{m}\left(Z_{m}^{n-1}\cdot p_{l}+c_{k}\right)\right]$$

To solve Equation (Equation 6) it can either fit the exponential term *n*, which it needs to use a non-linear least squares solver or to use a fixed n and solve it for *p* and *c* using a linear method.

In this case, we have used a fixed *n* = 3.0 and solved the minimization problem by using MATLAB’s function lsqnonlin (https://www.mathworks.com/help/optim/ug/lsqnonlin.html) (accessed on 10 January 2023).

Other proposed methods involved an initial nonlinear calibration, then, using for n parameter a fixed value determined by the first calibration to perform a second linear calibration to estimate *p* and *c*.

To illustrate the performances of our method, we have reconstructed the image of a reference sample (Figure 4a) using our algorithm (Figure 4b), Jumanzarov et al. [22] (Figure 4c), as well as van Aarle et al. [35] ones (Figure 4d) of which reconstructing time are presented in Table 3. The resulted CT images reproduced in Figure 4b–d show a significant resemblance. In spite of this, for a more objective characterization of their similarity, we have compared images histograms (Figure 5), by using more numerical tests, e.g., Spearman’s correlation coefficient ρ, Tuckey’s Q, Mann–Whitney as well as Dunnett post hoc tests. These results, reproduced in Table 4, point towards a significant resemblance between present work algorithm and the Jumanzarov et al. [22] one, as the image histograms reproduced in (Figure 5) show.

### 5.2. Effective Atomic Number and Density Calculation

Once the system calibrated, it was possible, by using the experimentally determinated LACs, values to calculate both Z_*eff*_ and ρ values by solving the following equation:(7)p1c1......p64c64μρ=μ1...μ64

This was performed by using a linear least squares solver to approximate the best solution for the system, in this case the MATLAB lsqnonneg (https://www.mathworks.com/help/optim/ug/lsqnonneg.html) (accessed on 10 January 2023) routine. Once the atomic number Z and density ρ are determined, the Z_*eff*_ values were calculated using the relation (8) derived from Equations (5) and (7):(8)$$Z_{eff}={\left(\frac{Z}{\rho }\right)}^{\frac{1}{n-1}}$$

### 5.3. Material Identification

For calibration, we used five materials with atomic numbers between 7.2 and 9.8 and densities between 0.61 and 1.45 g/cm^3^, e.g., PPH (Polypropylene), as well as four explosive simulants XM05X, XM06X, XM11GEX, and XM15X (https://www.officer.com/home/company/10031153/xm-nestt-div-of-van-aken) (accessed on 10 January 2023) (Table 2).

The proposed method of material identification presented and discussed before was validated by means of another five different materials, i.e., water, PA12 (polyamide 12 known as nylon 12), and explosive simulants XM03X, XM04X1, and XM08X of which certified as well as experimentally determined effective atomic numbers and densities (in g/cm^3^) are presented in Table 2. For all of them we have achieved a relative error of under 3% for both Z_*eff*_ and ρ values which means that, thus, experimentally determined parameters could be considered coincident with the certified one at *p* < 0.05.

The final results of concerning densities and effective atomic numbers of water, PA12, and explosive simulants XM03X, XM04X1, and XM08X are illustrated in Figure 6a,b. At can be remarked, a good identification with respect to provided data was noticed for all investigated materials, but when considered the reciprocal differences, XM023 and PA12 appear almost identical. At the same time, using ANOVA Student (same mean) and Wilcoxon (same median) tests, the best concordance between certified and determined values of PA12, and explosive simulants XM03X, XM04X1, and XM08X was attained for the effective atomic numbers (Table 4).

## 6. Conclusions

Given the final goal to develop a rapid and reliable procedure to identify presumed explosives or other sensitive materials based on multi-spectral computed tomography (MSCT), it was necessary to go through the following stages:To elaborate a technique to compensate for beam hardening and other non-linear effects which could natively influence the energy dependency of the Linear Attenuation Coefficient (LAC) between 20 and 160 keV, i.e., the energy domain currently utilized in the X-ray MSCT;To develop a 3D fast reconstruction algorithm able to furnish confident results concerning the local LAC values corresponding to each utilized energy bin, i.e., 64 values from 19.8 to 158.4 keV;To create a set of algorithms permitting a simultaneous determination of density and the affective atomic number of investigated materials.

In view of these commandments, it was possible to determine both density and effective atomic number of complex objects in less than 24 s, with accuracy and precision less than 3%, a significantly better performance with respect to the reported literature values.

## Figures and Tables

**Figure 1 materials-16-01654-f001:**
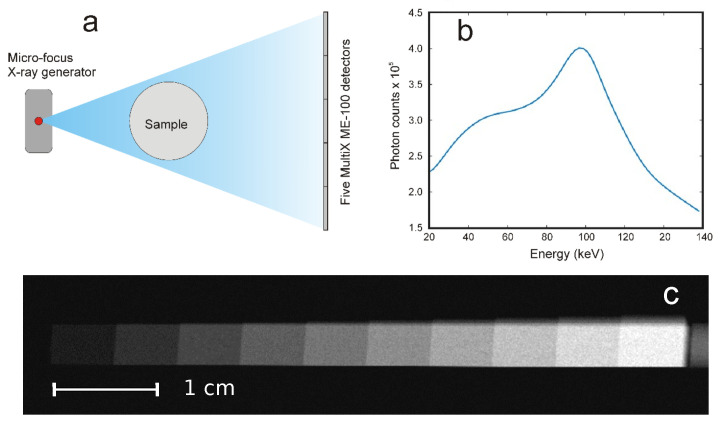
The CT setup (**a**), the experimental 83.6 keV X-ray spectrum as recorded by the MultiX ME-100TM detector (**b**) and a 83.6 keV experimental image of the PET test piece (**c**).

**Figure 2 materials-16-01654-f002:**
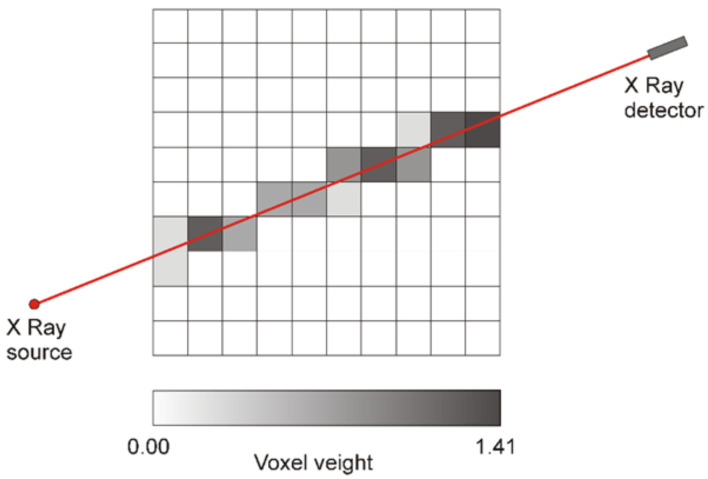
The simplified geometry of projects acquisition. The grey nuances can be correlated with the length of X-ray beam through each voxel.

**Figure 3 materials-16-01654-f003:**
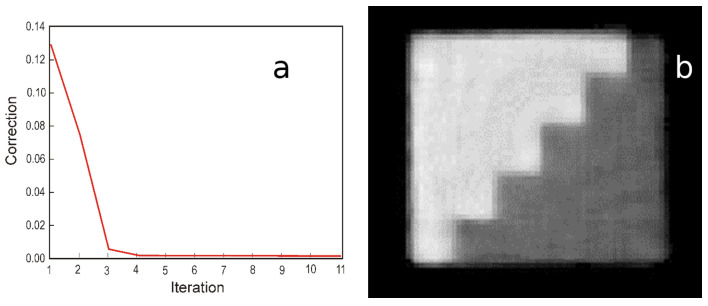
Average pixel change for each iteration, showing convergence after 11 steps (**a**) and the resulted image after 11 iteration of two joined objects evidencing a boundaries between materials (**b**).

**Figure 4 materials-16-01654-f004:**
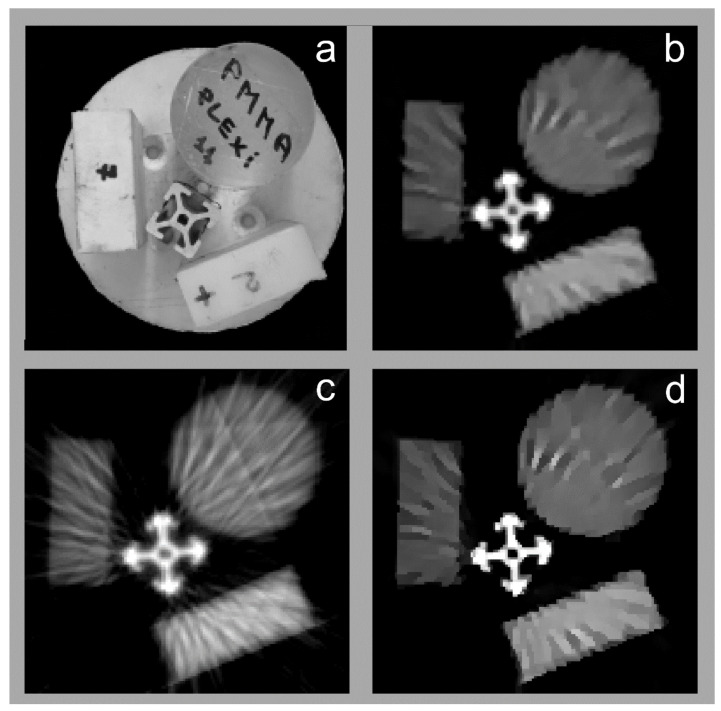
The sample object (**a**), as well as its reconstructed CT image using present work algorithm (**b**), as well as the Jumanzarov et al. [22] (**c**) and van Aarle et al. [35] ones (**d**).

**Figure 5 materials-16-01654-f005:**
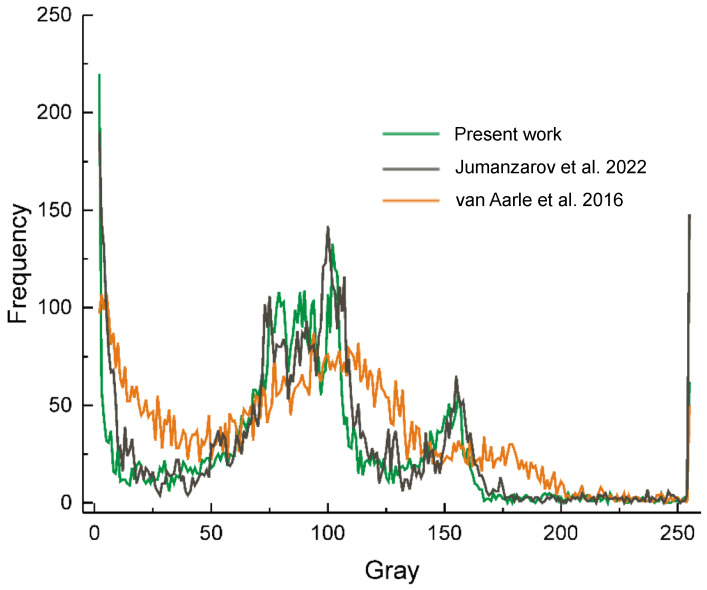
The histograms of CT images reproduced in Figure 4, e.g., present work algorithm (Figure 4b), Jumanzarov et al. [22] (Figure 4c) and van Aarle et al. [35] (Figure 4d).

**Figure 6 materials-16-01654-f006:**
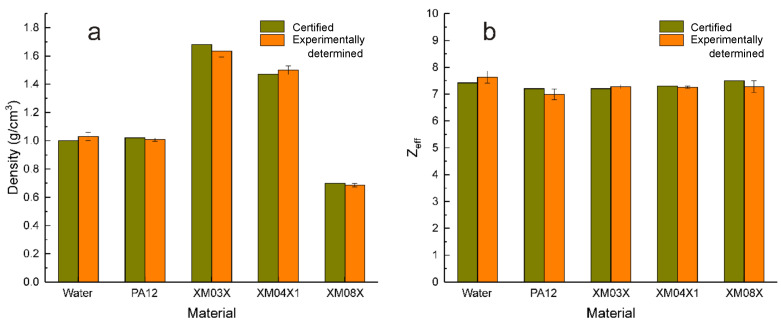
The certified and experimentally determined density values (**a**) and effective atomic number (**b**) of water, PA12 (polyamide 12 known as nylon 12), and explosive simulants XM03X, XM04X1, as well as XM08X.

**Table 1 materials-16-01654-t001:** Z_*eff*_ and ρ values of chosen CRM used for calibration.

CRM	Z_*eff*_	ρ (g/cm^3^)
PET	5.65	1.38
PTFE	7.30	1.45
PVC	9.20	1.37
PPH	7.60	0.91
Al	13	2.70

**Table 2 materials-16-01654-t002:** The execution metric for 40 runs which generated 64 images of 300 × 300 pixels.

Total Execution Time (s)	RAM Usage	Average CPU Usage
23.37	1.33 GB	88%

**Table 3 materials-16-01654-t003:** The Speraman’s correlation coefficient as well as the probabilities the reconstructed CT image to be similar to Jumanzarov et al. [22] and van Aarle et al. [35] consistent with Tukey’s Q, Mann–Whitney and Dunnett post hoc tests.

	Spearman’s ρ	Tukey’s *Q*	Mann-Whitney	Dunnett
Literature	Present Work	Present Work	Present Work	Present Work
[22]	0.87	0.53	0.59	0.67
[35]	0.76	0.00	0.1	0.01

**Table 4 materials-16-01654-t004:** The effective atomic number, density (in gcm^−3^) of the materials used for validation, as well as the probability, the experimentally, and the calibrated parameters to be the same according to Student *t* (same mean) and Wilcoxon (same median) tests. The relative error in determining both parameters is expressed in %.

Material	Z_*eff*_ Certified	Z_*eff*_ Experimental	Relative Error	Density Certified	Density Experimental	Relative Error
Water	7.42	7.64	2.94	1.00	1.03	2.98
PA12	7.20	6.99	−2.87	1.02	1.01	−1.11
XM03X	7.20	7.28	1.07	1.68	1.64	−2.65
XM04X1	7.30	7.26	−0.56	1.47	1.50	2.04
XM08X	7.50	7.28	−2.94	1.00	1.03	2.98
* **t** * **-test**	same mean	0.81		0.62		
**Wilcoxon test**	same median	0.87		0.81		

## Data Availability

Not applicable.

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
