# Peer review of "Multi-Energy and Fast-Convergence Iterative Reconstruction Algorithm for Organic Material Identification Using X-ray Computed Tomography"

_materials, 2023, doi:10.3390/ma16041654_

Round 1

Reviewer 1 Report

This is a very good first draft and exciting to read.  The paper ranks as a first draft due to a large number of typos.  

Line 299: should be average atomic number

Fig 8 X-axis label should be density

The field of multi-energy tomography is very rich.  The  paper should have more citations of the ongoing research

Author Response

Reviewer 1 (rd ink)

Remark

This is a very good first draft and exciting to read.  The paper ranks as a first draft due to a large number of typos.  

Answer

We have corrected as many as possible typos. All changes in the revised versions are edited with red, blue or green ink corresponding to the remarks of each reviewer

Remark

Line 299: should be average atomic number

Answer

We have corrected everywhere

Remark

Fig 7 X-axis label should be density

Answer

Figure 7 was totally removed as it repeated the in a different way the data illustrated in Fig. 6

Remark

The field of multi-energy tomography is very rich.  The  paper should have more citations of the ongoing research

Answer

We have included more references, about 17, marked on the revised text with red, blue and green ink.

Reviewer 2 Report

The paper introduces a specialized multi stage procedure, based on the multi-energy Computed Tomography investigation within 20-160 keV domain, to identify various organic material including explosives and narcotics. The approach is valid after experimental verification.

The main concerns are listed as follow:

1. In the third paragraph, how to divide the large domain of energies into numerous bins? I recommend you can introduce the principle of the Multi-Spectral Computed Tomography (MSCT) in more detail.

2. What are the current methods for detecting organic material including explosives and narcotics? I recommend you can make a brief introduction.

3. In the fourth paragraph, does multiple energies refer to multiple energy bins? If not, what does multiple energy mean?

4. In the fourth paragraph, how to judge whether there are explosives or narcotics according to the density and atomic effective number of the controlled items?

5. In the seventh paragraph, what is the number of projections? How does the number of projections affect this method?

6. Some deep learning methods are missing. [1]

Graph-based Few-Shot Learning with Transformed Feature Propagation and Optimal Class Allocation

 [2] 

Deep-IRTarget: An Automatic Target Detector in Infrared Imagery using Dual-domain Feature Extraction and Allocation

Author Response

Reviewer 2 (blue ink)

The paper introduces a specialized multi stage procedure, based on the multi-energy Computed Tomography investigation within 20-160 keV domain, to identify various organic material including explosives and narcotics. The approach is valid after experimental verification.

The main concerns are listed as follow:

Remark

1a. In the third paragraph, how to divide the large domain of energies into numerous bins?

Answer

We have explained (row 101-102): configured by manufacturer with 64 equal-energy bins (blue ink)

Remark

1b. I recommend you can introduce the principle of the Multi-Spectral Computed Tomography (MSCT) in more detail.

Answer

In the introduction, we have explained in more details the advantages of MSCT (row 45-48, blue ink).

Remark

2. What are the current methods for detecting organic material including explosives and narcotics? I recommend you can make a brief introduction.

Answer

In the introduction, we have included a new paragraph (row 19-26) (red ink)

Remark

3. In the fourth paragraph, does multiple energies refer to multiple energy bins? If not, what does multiple energy mean?

Answer

We have explained that each energy bin corresponds to a different energy interval (row 101-102) blue ink

Remark

4. In the fourth paragraph, how to judge whether there are explosives or narcotics according to the density and atomic effective number of the controlled items?

Answer

We have mentioned in the Introduction (row 28-33) than only a simultaneous determination of the effective atomic number and density permit a differentiation between explosives and the other organic matter. Of course, this process in not absolute, but, as mentioned in Introduction (row 37-39 ), This screening it is not without errors, but the reason is to reduce their number, i.e. false positive or false negative results to an absolute minimum (blue ink)

Remark

5. In the seventh paragraph, what is the number of projections? How does the number of projections affect this method?

Answer

We have included a short explanation concerning the chosen number of projections (row 108-110) (blue ink)

Remark

6. Some deep learning methods are missing.

Answer

We have include a small introductory paragraph (row 55-61) as well as the corresponding references explaining the importance of deep learning method. In our case, as son as the number of investigated samples will reach a critical number, we would use the AI to solve this problem, Only 5-7 samples are, in our opinion, totally insufficient to train a neural network. Besides it, the main goal of our project was to significantly decrease the computing time without losses of accuracy and precision.

Graph-based Few-Shot Learning with Transformed Feature Propagation and Optimal Class Allocation

 [2] 

Deep-IRTarget: An Automatic Target Detector in Infrared Imagery using Dual-domain Feature Extraction and Allocation

Reviewer 3 Report

This is a fundamental work on proposing an algorithm for material identification using X CT. The x-ray technique always suffers the difficulty to identify different materials especially with similar densities. In this study, a specialized multi stage procedure based on the multi-energy computed tomography investigation was established. The performance was claimed to determine the density and effective atomic number of the materials in about 24 s with an accuracy less than 3%. The work is within the scope of the journal. The results are well supported by the data. The following questions should be answered before further consideration:

1.      Scale bars should be added to all photo and CT images.

2.      It seems that there existed significant artifacts in the reconstructed images in Fig. 5. How will this affect the results of the density determination

3.      Besides the accuracy in density identification, the quality of the image should also be discussed as describing the shape of the objectives is the fundamental function of CT.

4.      Are the data used in Fig. 7 and 8 the same? If so, not necessary to show them twice.

5.      The introduction is not so comprehensive and some very recent work using X-ray CT reconstruction can be introduced: Applied Energy, 2023, 331: 120372

6.      Some language errors, e.g. “with an accuracy …” in the abstract; “veight” in Fig. 3; line 224 “As mentioned” and many other places. Please check throughout the text.

Author Response

Reviewer 3 (green ink)

This is a fundamental work on proposing an algorithm for material identification using X CT. The x-ray technique always suffers the difficulty to identify different materials especially with similar densities. In this study, a specialized multi stage procedure based on the multi-energy computed tomography investigation was established. The performance was claimed to determine the density and effective atomic number of the materials in about 24 s with an accuracy less than 3%. The work is within the scope of the journal. The results are well supported by the data. The following questions should be answered before further consideration:

Remark

1. Scale bars should be added to all photo and CT images.

Answer

We have added to all images, i.e. Fig. 1c, Fig. 3b and Fig. 4c to 4d

Remark

2.It seems that there existed significant artifacts in the reconstructed images in Fig. 5. How will this affect the results of the density determination

Answer

As we have mentioned, it was a trade off between the reconstructed CT images and acquisition time, i.e. number of projections (row 109-111). The actual number of projections equal to 36 was the minimum which permitted a the relative errors in determining both Zeff and density of considered explosive stimulants to be less than 3%. Of course, by increasing the projection number, the CT images would have a better spatial resolution, but with the price of acquisition time.

Remark

3.Besides the accuracy in density identification, the quality of the image should also be discussed as describing the shape of the objectives is the fundamental function of CT.

Answer

It is perfectly true, but in our case we were interesting in a simultaneous determination of both both Zeff and density with relative errors less then 3%, as previous studies reported. For instance, the images reproduced in Fig. 4c and 4d have the same quality as our images illustrated by Fig. 4b.

Remark

4.Are the data used in Fig. 7 and 8 the same? If so, not necessary to show them twice.

Answer

We have removed fig. 7 as it repeats in essence the fig. 6

Remark

5.The introduction is not so comprehensive and some very recent work using X-ray CT reconstruction can be introduced: Applied Energy, 2023, 331: 120372

Answer

We have included more up-to date references mainly marked with green ink

Remark

6. Some language errors, e.g. “with an accuracy …” in the abstract; “veight” in Fig. 3; line 224 “As mentioned” and many other places. Please check throughout the text.

Answer

We have thoroughly checked entire manuscript correcting any misprints observed

Round 2

Reviewer 1 Report

The introduction is improved, but there are no changes in the other sections.  The original review still applies:  very interesting work, but presented as a draft (not complete) manuscript.

Reviewer 2 Report

Accept